# Refocusing of Multiple Moving Targets Based on the Joint Sparse Processing of One Channel Synthetic Aperture Radar Imagery Patches

**Xin Wang * and Ling Qiao**

College of Telecommunations and Information Engineering, Nanjing University of Posts and Telecommunications, Nanjing 210023, China; jolin970921@163.com
* Correspondence: wangx@njupt.edu.cn; Tel.: +86-13913312604

**Abstract:** A sparse-based refocusing methodology for multiple slow-moving targets (MTs) located inside strong clutter regions is proposed in this paper. The defocused regions of MTs in synthetic aperture radar (SAR) imagery were utilized here instead of the whole original radar data. A joint radar projection operator for the static and moving objects was formulated and employed to construct an optimization problem. The Lp norm constraint was utilized to promote the separation of MT data and the suppression of clutter. After the joint sparse imaging processing, the energy of strong static targets could be suppressed significantly in the reconstructed MT imagery. The static scene imagery could be derived simultaneously without the defocused MT. Finally, numerical simulations were used verify the validity and robustness of the proposed methodology.

**Keywords:** SAR moving target imaging; joint sparse imaging; time frequency analysis

## 1. Introduction

Synthetic aperture radar (SAR) [1], which is an advanced remote sensing system, has been widely used in the past few decades. Taking advantage of the two-dimensional high-resolution capability, the SAR system can implement accurate target classification, recognition, and location finding after imaging processing. However, when there are multiple moving targets (MTs) in the illuminated scenery, smearing and geometry position deviation [2,3] generally emerge in the constructed imagery. Ground moving target imaging (GMTIm) thus becomes very important and has obtained more and more interest in recent years.

The smearing and geometry distortion of MTs result from the lack of prior knowledge of moving velocities. Therefore, most of the papers on MT imaging focus on the estimation of the moving parameters [4–6] and the designing of filters for motion phase terms compensation [7–10]. In these papers, the Doppler parameters were estimated using time-frequency analysis or other methodologies. Then, one-dimensional or two-dimensional filters were developed to implement MT imaging. The original SAR data, in which the clutter and MT radar data are mixed and are difficult to separate, is generally needed in the above algorithms. The focal quality of the derived MT imagery using the above algorithms will thus be affected by the existence of strong clutter.

Instead of using the original SAR data, many scholars proposed to implement MT refocusing based on the defocused regions of interest (ROI), where most of the clutter is separated using a filtering operation. The analytical expressions of the phase error terms of MTs in the wavenumber domain are derived in References [11–13]. Then, refocusing operation is implemented via motion phase error compensation and a Fourier transform. Some autofocus-based methods [14,15] have also been discussed to realize MT refocusing by searching for moving parameters in a predefined region. The

focal quality of the derived imagery is measured to find the best result. However, a filtering operation generally cannot remove the energy of clutter and static targets completely, especially when MTs are located inside a strong clutter region. These algorithms cannot be applied for multiple MTs imaging simultaneously and the computational burden is generally severe.

When there are strong static targets located inside ROIs, the performance of the above algorithms will degrade quickly since the defocused MTs are difficult to extract using a filtering operation. Some residual clutter will still exist and be smeared in the reconstructed MT imagery. To solve this problem, many papers have proposed suppression of the clutter before MT refocusing in multi-channel SAR. An over-completed velocity dictionary and a Doppler dictionary are constructed to realize the refocusing of multiple MTs [16–18] after the displaced phase center antenna (DPCA) processing. The norm regularization is used to constrain the solution to be sparse. Furthermore, a range frequency reversal transform-fractional Fourier transform (RFRT-FrFT) [19] has recently been developed for MT range cell migration correction (RCMC) after DPCA. However, they had to consider all the possible values of moving parameters in Rodrigo and Wang [17] and the performance of the RFRT-FrFT method will be affected by the existence of cross-terms and strong clutter. These above algorithms were all developed based on multi-channel data, which is not suitable for one-channel SAR where the system freedom is reduced. The non-coherent subtraction, along-track interferometry (ATI) [20], and space-time adaptive processing (STAP) based on virtual multiple-channels [21] have been discussed to remove the clutter from one-channel SAR data. However, the performance of non-coherent subtraction and ATI will degrade in low signal to clutter ratio (SCR) cases and the STAP method needs a high pulse repetition frequency.

To implement the MTs' refocusing processing in one-channel SAR, we propose a joint sparse-based method in this paper. The Doppler characteristic differences between the MT signal and clutter data, which were used for detecting moving objects [22,23], were utilized in this paper to formulate two different projection operators. The data from MTs and the static scene, which are mixed in the defocused ROIs, will be projected to different positions according to its relevance with system functions. To promote the separation of data and the suppression of artifacts and side lobes, we consider employing a sparse constraint on the solution. This has also been utilized in compressed sensing (CS) [24,25] to realize SAR or inverse synthetic aperture radar (ISAR) imaging [26–30], tomography, and ground moving target indication in the past few decades. The CS theory can implement the complete recovery of the original signal with fewer measurements than the Nyquist sampling rate and thus is very useful when the raw data is undersampling or a part is missing. In Patel et al. [26], the CS theory is discussed to reconstruct the imagery with very few sampling data. Then, Hu et al. investigated a series of CS-based algorithms from different aspects [27–30] to implement high-resolution imaging based on data received from sparse apertures or random down-sampling. Moreover, the sparsity of the solution has also been utilized and the CS method was extended to tomography processing [31–33] and MT indication [33,34], where various iteration computation methodologies were developed.

In view of the above attractive and successful applications, we try to use the same sparse constraint in CS to realize the reconstruction of a static scene and MT imageries and the suppression of clutter in this paper. After the joint sparse-based refocusing, MT imagery and static scene imagery could be derived. First, the general SAR data collection geometry and signal model are described in Section 2. Then, the joint sparse imaging methodology is presented and discussed in Section 3. Finally, numerical simulation verifies that the algorithm can implement multiple moving target imaging conveniently.

## 2. Signal Model

Figure 1 depicts a SAR data collection geometry, where the scene center is defined as the origin of the coordinate system. The transmitter and receiver are placed on the same platform that flies with a constant altitude $h_t$ and velocity $v_t$. The illuminated scene of interest is static during the data collection integration time while one MT located at $(x_m, y_m)$ moves with constant velocities. The instantaneous slant range for an MT is denoted as $R_m(t)$ at an arbitrary azimuth sampling time $t$.

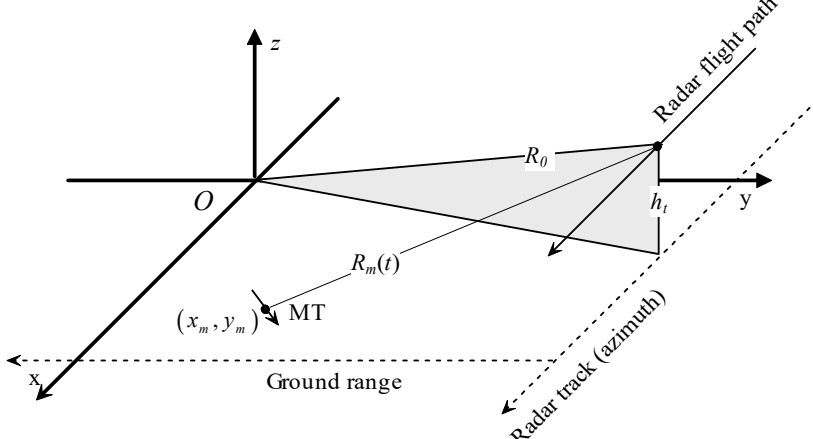

**Figure 1.** SAR data collection geometry.

Radar back data on the receiver is a mixture of the reflected signals from all the targets in the illuminated scene. Assuming radar transmits a linear frequency modulation (LFM) signal, SAR data after range compression can be formulated as:

$$Y = C + F_m P T_m + n_0 \tag{1}$$

where, $Y \in \mathbb{C}^{N_d \times 1}$ is the complex column vector derived by stacking the range compressed SAR data $\mathbb{C}^{N_d \times 1}$ is the complex vector space where the dimension of vector is described by the superscript, $C \in \mathbb{C}^{N_I \times 1}$ is referred to as clutter, $F_m \in \mathbb{C}^{N_d \times N_I}$ is the radar projection operator of MT, and the notations $N_d$ and $N_I$ refer to the length of radar data and imagery, respectively. In Equation (1), $T_m \in \mathbb{C}^{N_I \times 1}$ is a column vector corresponding to the radar cross-section (RCS) of MTs and $n_0$ is the noise vector.

## 3. MT Refocusing Methodology

It can be seen from Equation (1) that SAR received data is a collection of the clutter and MT signal from the illuminated scene. To derive the MT imagery with a high focal quality, we should separate or suppress the clutter in advance. However, this is difficult to realize when the smeared MT overlaps with the static targets in the imagery domain. The energy of the clutter cannot be suppressed completely by the traditional suppression methods. Utilizing the different Doppler characteristics of radar data from the static and moving objects, we describe the conversion of the MT refocusing into a sparse optimization problem in this section. The static scene and MT imageries could be derived simultaneously after the iteration computation, which is described in detail in the following discussion.

### 3.1. Clutter and MT Data Separation

The SAR imagery with defocused MTs that was derived after imaging processing could be obtained using:

$$\begin{aligned} S &= F_I(C + F_m T_m + n_0) \\ &= F_s T + n_I \end{aligned} \tag{2}$$

where, $F_s = [F_I F, F_I F_m]$

In Equation (2), $S \in \mathbb{C}^{N_I \times 1}$ is the stacked vector of SAR imagery and $F \in \mathbb{C}^{N_d \times N_I}$ is the radar projection operator of the static targets in which a column vector is a stacked vector of radar data from a static target. The notation $F_I \in \mathbb{C}^{N_I \times N_d}$ in Equation (2) is an inverse imaging matrix; the column vector $T = [T_s, T_m]^T \in \mathbb{C}^{2N_I \times 1}$ consists of a static scene imagery $T_s$ and the defocused MT area $T_m$, where the element $T_i$ can be referred to as radar cross section (RCS) information; and $n_I$ refers to the noise in the SAR imagery domain.

Here, a joint imaging operator $F_s$ that incorporates the SAR system model of static and moving targets and the inverse imaging operator was employed. Generally speaking, $F_I$ varies with SAR processing algorithms. Here, $F_I = F^H$, which means the back projection algorithm was utilized to guarantee the accuracy of the system model, where the subscript H denotes the conjugate transpose of a matrix.

Due to the lack of prior information regarding the MTs, the exact analytical expression of $F_m$ is generally unknown. To reconstruct the separated static scene and MT imageries with a high quality, we built the following optimization problem using:

$$\min_{T,F_m} J(T,F_m) = \min_{T,F_m} \|S - F_s T\|_2^2 + \lambda_1 \|T\|_p^p \tag{3}$$

where $\|\cdot\|_p^p$ denotes the $\ell_p$ norm and $\lambda_1$ is a positive scaling parameter. In Equation (3), the first term is a data fidelity term, which incorporates the joint SAR observation model in Equation (2) and the RCS information of the illuminated targets. The adoption of the data fidelity term aims to implement the inversion of radar back data. In practice, the Doppler characteristics of radar back data from MTs are very different from that of a static scene. Thus, the mix-received radar data will be projected on the joint system operator and separated to generate the static scene imagery $T_c$ and the MT imagery $T_m$. Then, the smeared clutter and static targets exist in the MT imagery with low amplitude values, which is the same as that in the static scene imagery. These smeared impulse response functions will be viewed as artifacts and can generally be suppressed using the sparse constraint. Hence, though the static scene might not be sparse in practice, the sparse constraint can still promote the separation and suppression of clutter, artifacts, and side lobe in the reconstructed imageries, which was verified by the simulation results in Section 4.

To constrain the solution of an optimization problem to be sparse, the $\ell_p$ norm with $p = 0$ was usually selected. In this case, Equation (3) became an $\ell_0$ regularization problem, which is NP-hard. Greedy methodologies [23] have been developed to solve this problem approximately. However, the accuracy of these methods will degrade when the moving parameter estimation error increases in low SCR or long integration aperture cases. Recently, the $\ell_1$ norm has been widely used instead. However, as discussed in References [28,29], the $\ell_p$ norm with $0 < p < 1$ will result in a more sparse solution in comparison, and thus was used in our method.

Another factor that affects the separation of MT is the design of the radar operator $F_m$. In practical data processing, the initialized $\widehat{F}_m^{\,0}$ could be formulated based on some prior information or the estimation results of moving parameters. When the mismatched phase error of $\widehat{F}_m^{\,0}$ is not too large, a static scene and MT imageries could be separated and derived directly. Otherwise, the residual energy of the clutter might still exist in the reconstructed MT imagery. To avoid this problem, we tried to update the operator $F_m$ during the iterative computation interval until the solution converged. The detailed iterative computation is described in next section.

*3.2. Iterative Solution*

The iterative computation of Equation (3) can be divided into two sub-problems. First, updating T by fixing $F_m$ and then updating $F_m$ by fixing T. The two procedures were iteratively implemented until the derived results converged.

Since the $\ell_p$ norm is non-differentiable around the origin, an exact solution of Equation (3) is difficult to obtain directly. As discussed in Onhon and Cetin [28], the following approximation expression was applied instead of the $\ell_p$ norm:

$$\|z\|_p^p \approx \sum_{i=1}^{N_z} \left(|z(i)|^2 + \varepsilon\right)^{p/2} \tag{4}$$

where $\varepsilon \geq 0$ is a small positive constant, $N_z$ denotes the length of the vector z, and $z(i)$ refers to the *i*th element in z. By substituting Equation (4) into the cost function in Equation (3), a modified equation is given as:

$$J_m(\mathrm{T},\mathrm{F}_m) = \|\mathrm{S} - \mathrm{F}_I\mathrm{T}\|_2^2 + \lambda_1\sum_{i=1}^{2N_I}\left(|\mathrm{T}(i)|^2 + \varepsilon\right)^{p/2} \tag{5}$$

The modified cost function $J_m(\mathrm{T},\mathrm{F}_m)$ will always be close to $J(\mathrm{T},\mathrm{F}_m)$ when $\varepsilon \to 0$. There is no closed-form solution for the minimization of Equation (5) and the quasi-Newton methods may be used to derive the solution as discussed in Cetin and Karl [29].

Calculating the gradient of Equation (5) to the real part and imaginary part of T, we can obtain the following iterative formula based on the Hessian matrix approximation:

$$\widehat{\mathrm{T}}^{(n+1)} = \widehat{\mathrm{T}}^{(n)} - \gamma\left[\mathrm{H}\left(\widehat{\mathrm{T}}^{(n)}\right)\right]^{-1}\nabla J_m\left(\widehat{\mathrm{T}}^{(n)},\mathrm{F}_m\right) \tag{6}$$

where

$$\mathrm{H}\left(\widehat{\mathrm{T}}^{(n)}\right) \triangleq 2\mathrm{F}_s^{\mathrm{H}}\mathrm{F}_s + p\lambda_1 diag\left\{\left(\left|\widehat{\mathrm{T}}^{(n)}(j)\right|^2 + \varepsilon\right)^{p/2-1}\right\} \tag{7.a}$$

$$\nabla J_m\left(\widehat{\mathrm{T}}^{(n)},\mathrm{F}_m\right) = \mathrm{H}\left(\widehat{\mathrm{T}}^{(n)}\right)\widehat{\mathrm{T}}^{(n)} - 2\mathrm{F}_s^{\mathrm{H}}\mathrm{S} \tag{7.b}$$

In Equation (7), $\gamma$ denotes the iteration step, $\nabla J_m(\cdot)$ is the complex gradient of the cost function $J_m(\mathrm{T})$, $\mathrm{T}^{(n)}$ is the estimation result after the *n*th iteration, and $\left[\mathrm{H}\left(\widehat{\mathrm{T}}^{(n)}\right)\right]^{-1}$ is the inversion matrix of $\mathrm{H}\left(\widehat{\mathrm{T}}^{(n)}\right)$. Substituting the expression for the gradient into Equation (6), the iterative formula is rewritten as:

$$\left[\mathrm{H}\left(\widehat{\mathrm{T}}^{(n)}\right)\right]\widehat{\mathrm{T}}^{(n+1)} = (1 - \gamma)\left[\mathrm{H}\left(\widehat{\mathrm{T}}^{(n)}\right)\right]\widehat{\mathrm{T}}^{(n)} + 2\gamma\mathrm{F}_I^{\mathrm{H}}\mathrm{S} \tag{8}$$

Equation (8) is a linear equation with conjugate matrix coefficients. The sparsity of $\mathrm{H}\left(\widehat{\mathrm{T}}_i^{(n)}\right)$ is increased by neglecting elements in $\mathrm{F}_I^{\mathrm{H}}\mathrm{F}_I$ whose magnitudes are smaller than 1% of the largest element. Hence, the conjugate gradient method could be applied to search for the solution of Equation (8).

The second iteration computation is done by fixing T to solve for $\mathrm{F}_m$. The optimization problem then becomes:

$$\min_{\mathrm{F}_m}\left\|\mathrm{S} - \mathrm{F}^{\mathrm{H}}\mathrm{F}\widehat{\mathrm{T}}_s^{(n)} - \mathrm{F}^{\mathrm{H}}\mathrm{F}_m\widehat{\mathrm{T}}_m^{(n)}\right\|_2^2 + \lambda_1\left\|\widehat{\mathrm{T}}^{(n)}\right\|_p^p \tag{9}$$

As no additional constraint is performed on $\mathrm{F}_m$ and the phase information is not retained after the iteration computation, the direct solution of Equation (9) is difficult to obtain. In References [13,14], the MT parameters are updated by searching in a predefined region based on the maximization of imagery sharpness or entropy. However, this method is computationally expensive and cannot cope with multiple MTs with different velocities. Herein, MT patch data was reconstructed and moving parameters were estimated, which was expressed as:

$$\widehat{\mathrm{Y}}_m^{(n+1)} = \widehat{\mathrm{F}}_m^{(n)}\left[\left|\widehat{\mathrm{T}}_m^{(n)}\right|\odot\exp\left(\angle\widehat{\mathrm{T}}_m^{(0)}\right)\right] \tag{10}$$

where, $\odot$ denotes an elementwise product operation, $|\cdot|$ and $\angle\cdot$ indicate the computation of the amplitude and angle of a vector, and $\widehat{\mathrm{T}}_m^{(n)}$ and $\widehat{\mathrm{T}}_m^{(0)}$ refer to the constructed and initialized MT imagery,

respectively. Though the MT system operator $\widehat{F}_m^{(n)}$ might deviate from the real values, most of the clutter energy is still suppressed in the derived sparse MT imagery. Compared with the original SAR data, the SCR in the reconstructed MT data $\widehat{Y}_m$ will be decreased significantly. Thus, a parameter estimation operation could be performed on $\widehat{Y}_m$ directly to update the system function. The time-frequency analysis, interferometry, and sub-image subtraction methodologies have been developed for moving parameter estimation in one-channel SAR. Herein, the fractional Fourier transform after a keystone transform in Li et al. [35,36] combined with RCMC was utilized. Then, the next iterative processing could be carried out continually until the solution converged. The detailed iteration processing is listed in Algorithm 1.

---

**Algorithm 1:** Joint sparse-based MT refocusing

---

1. Input: Defocused imagery area of MT;
2. Initialization: $\widehat{T}^{(0)}$, $\widehat{F}_m^{(0)}$, F, $\lambda_1$, $\gamma$, $p$;
3. While not converged, do:
4. 　　　Update the reconstructed result $\widehat{T}^{(n+1)}$ according to Equation (8);
5. 　　　Update the MT projection operator $\widehat{F}_m^{(n+1)}$;
6. End
7. Output: Static scene imagery and MT imagery.

---

### 3.3. Implementation Issues

SAR imagery derived using back projection can be used as the initialization $\widehat{T}^{(0)}$. Assuming the estimated velocities of MTs along the $x$ and $y$ axes are $v_x' = v_x + \Delta v_x$ and $v_y' = v_y + \Delta v_y$, respectively, and an initial estimation of the radar MT projection operator $\widehat{F}_m^{(0)}$ could thus be constructed. The notations $v_x$ and $v_y$ are real velocities of the MT along the $x$ and $y$ axes, respectively, where $\Delta v_x$ and $\Delta v_y$ refer to mismatched velocity errors. In practice, if some prior information about MT is given in advance, a larger velocity could also be selected to start for the first iteration without parameter estimation.

Furthermore, since the elements in the re-stacked vector of SAR imagery and projection operator matrix are complex, the above iterative computation cannot be applied directly. One of the most widely used methods for solving the complex optimization problem is decomposing the complex matrix product into real matrix operations [28], which is also used in our paper. When the converging condition is satisfied, the static scene imagery and MT imagery will be obtained.

Moreover, the huge dimension of the SAR real data and imagery for the illuminated scene will increase the complexity of our method and limit the application. As the defocused area of MTs was usually distributed in small patches of SAR imagery, we did not need to construct the radar projection operator of the whole scene. Patch processing was introduced and employed in the proposed method to reduce the computational burden.

The defocused MT patches in the SAR imagery was extracted first as the input of iterative computation. Then, the SAR joint imaging operator $\widehat{F}_s$ corresponding to the imagery patches was formulated and utilized. The number of multiplication operations in the computation of $\widehat{F}_s$ was proportional to the dimension of SAR data and patch imagery. To further reduce the computational cost, we employed an azimuth down-sampling operation when reconstructing the radar projection operators and MT data $\widehat{Y}_m^{(n+1)}$. In iteration processing, a preconditioned conjugate gradient algorithm with high convergence speed is used and the computational burden is proportional to the length

of the SAR imagery vector T. As a result of the utilization of sub-patch processing and azimuth down-sampling, the computational cost in joint sparse MT imaging is decreased significantly.

In practical data processing, the size of one MT patch is generally very small (the patch size for one MT might be restricted to $30 \times 30$ in a low resolution case) and the number of iteration computations is less than 10 when the derived solution converges. Moreover, as the dimensions of the SAR imagery patches are very small, the computation of $F_s$ can be implemented approximately, where the value of the elements $F_s(i, j)$ with $|i - j| = m$ is the same. In this case, the total computational burden was much lower than that of autofocus processing of SAR data. In conclusion, the detailed flow of joint sparse imaging processing is depicted in Figure 2. After the iteration converged, the static scene and MT imageries could be derived simultaneously.

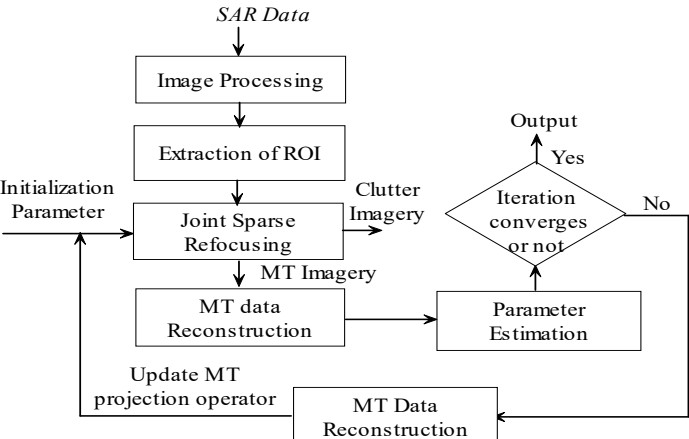

**Figure 2.** Joint sparse refocusing flow.

## 4. Simulation

In this section, numerical simulations are described to verify the proposed joint sparse-based algorithm, where the parameters are given in Table 1. Radar back data from five ground MTs, which are marked as MT1–5 in Figure 3a, and 72 static targets were simulated. The velocities and positions of the simulated MTs are listed in Table 2. The range-compressed MTs and SAR data are given in Figure 3b,c, respectively. It can be seen from these figures that the received radar data of MTs was masked by the clutter, which should be suppressed in advance.

**Table 1.** Simulation parameters.

| System Parameter | Numerical Value |
| --- | --- |
| Carrier frequency | 10 GHz |
| Range bandwidth | 300 MHz |
| Resolution in the x-axis | 0.41 m |
| Resolution in the y-axis | 0.23 m |
| Range resolution | 0.12 m |
| Azimuth resolution | 0.31 m |

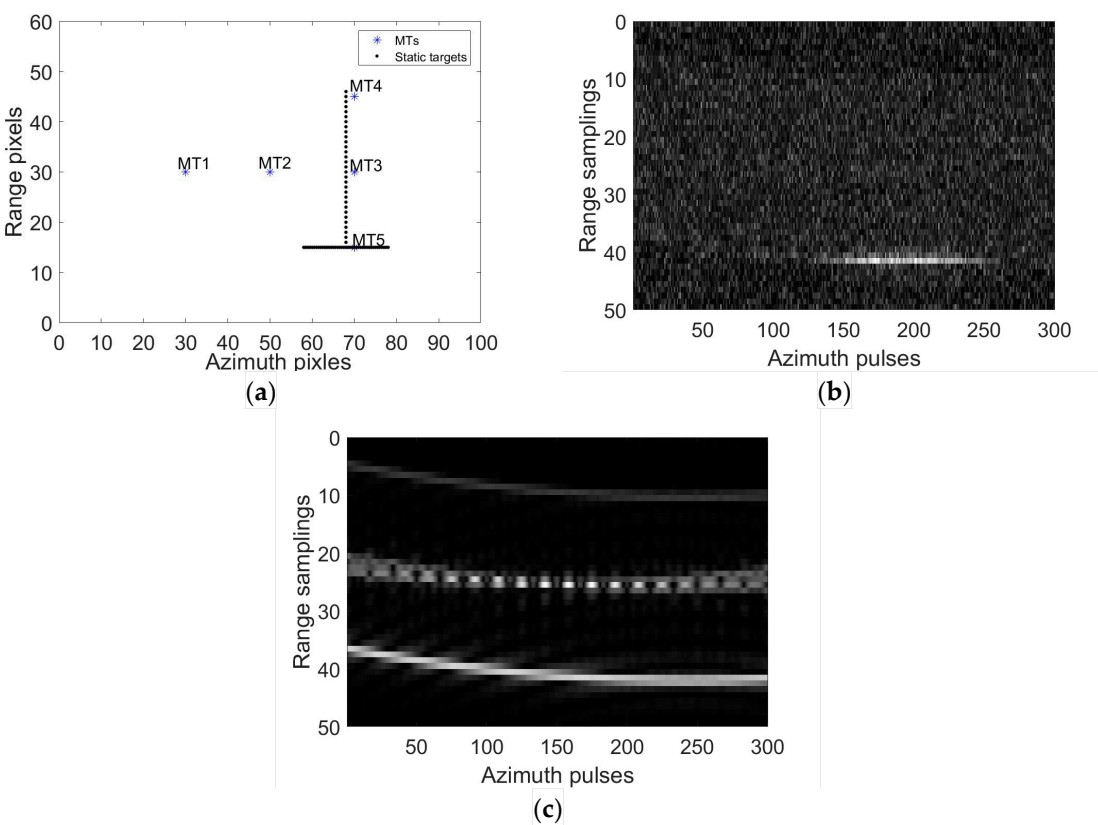

**Figure 3.** The simulated MTs and SAR data. (**a**) the geometry relationship of the MTs, (**b**) the range-compressed SAR data, and (**c**) the range-compressed MT radar data.

**Table 2.** Simulation parameters of the MTs.

|                              | **MT1**    | **MT2**    | **MT3**   | **MT4**     | **MT5**    |
| ---------------------------- | ---------- | ---------- | --------- | ----------- | ---------- |
| Azimuth position (pixel)     | 30         | 50         | 70        | 70          | 70         |
| Range position (pixel)       | 30         | 30         | 30        | 15          | 45         |
| $v_x$                        | 23 m/s     | 23 m/s     | 20 m/s    | 18 m/s      | 18 m/s     |
| $v_y$                        | 0.5 m/s    | 0.5 m/s    | 0.5 m/s   | 0.8 m/s     | 0.8 m/s    |
| SCR                          | −1.1 dB    | −1.1 dB    | −1.1 dB   | −11.2 dB    | −9.3 dB    |

After back-projection processing, the SAR images could be derived as in Figure 4a,b. In Figure 4b, the five ellipses indicate the locations of MTs that were smeared in the imagery. Actually, the smeared MTs overlapped with the strong static scene and were masked by the clutter. The SCR values were calculated as the ratio of the energies of MT and clutter in the SAR imagery and listed in Table 2. It could be seen from Figure 4b and Table 2 that the energy of MT4 and MT5 was much lower than that of the clutter. In this case, the clutter was difficult to suppress or separate to derive a result with high SCR.

Here, for comparison with our method, the matched filtering, interferometry method, and L1 norm regularization methods were performed on the defocused MT SAR imagery, and the results are given in Figure 4c–e. It can be seen from these figures that the clutter was suppressed to some extent but there was still residual energy for strong static targets, which affected the performance of MT detection, parameter estimation, and the imaging process. By contrast, the images of five MTs and 72 static targets were obtained with a high quality by using our joint sparse processing and the results are depicted in Figure 4f,g.

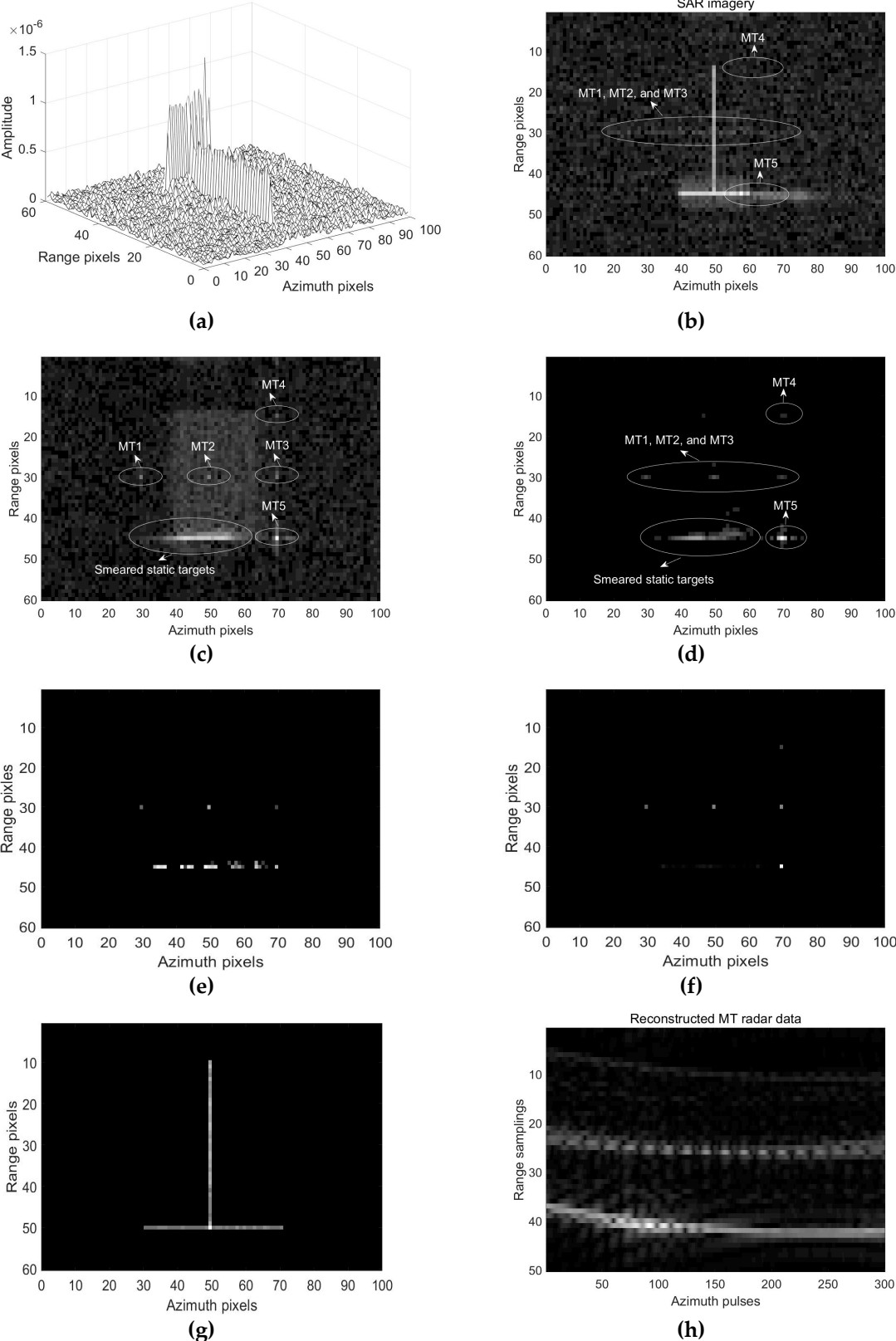

**Figure 4.** MF and sparse imaging results. (**a**) 3-D figure of SAR imagery; (**b**) 2-D SAR imagery; (**c**) imagery result derived by frequency domain matched filtering; (**d**) imagery result after interferometry; (**e**) imagery derived based on the L1 norm regularization; (**f**) and (**g**) MT imagery and static scene imagery, respectively, after the joint sparse processing; and (**h**) reconstructed MT radar data.

The initialized velocities $v'_x$ and $v'_y$, which were used in the first iteration, were taken to be 25 m/s and 0 m/s, respectively. The regularization parameters employed in the simulation were defined as $\lambda_1 = 0.3$ and $p = 0.5$. Though the initialized MT radar projection operator is mismatched, most of the cluttered energy was still suppressed in the derived imagery after the iteration computation, as shown in Figure 5b. MT radar patch data was then reconstructed based on Equation (9) and is given in Figure 5c. The Doppler parameter estimation operation was then performed after a keystone transform and range cell migration correction. The fractional Fourier transform was used and the estimation results of MT1 and MT5 are given in Figure 5e,f. We can find from these figures that the Doppler rate estimation results were very close to the real values. After updating the MT system operator based on the estimated parameters, we finally obtained the MT imagery in Figure 4f and radar patch data in Figure 4g.

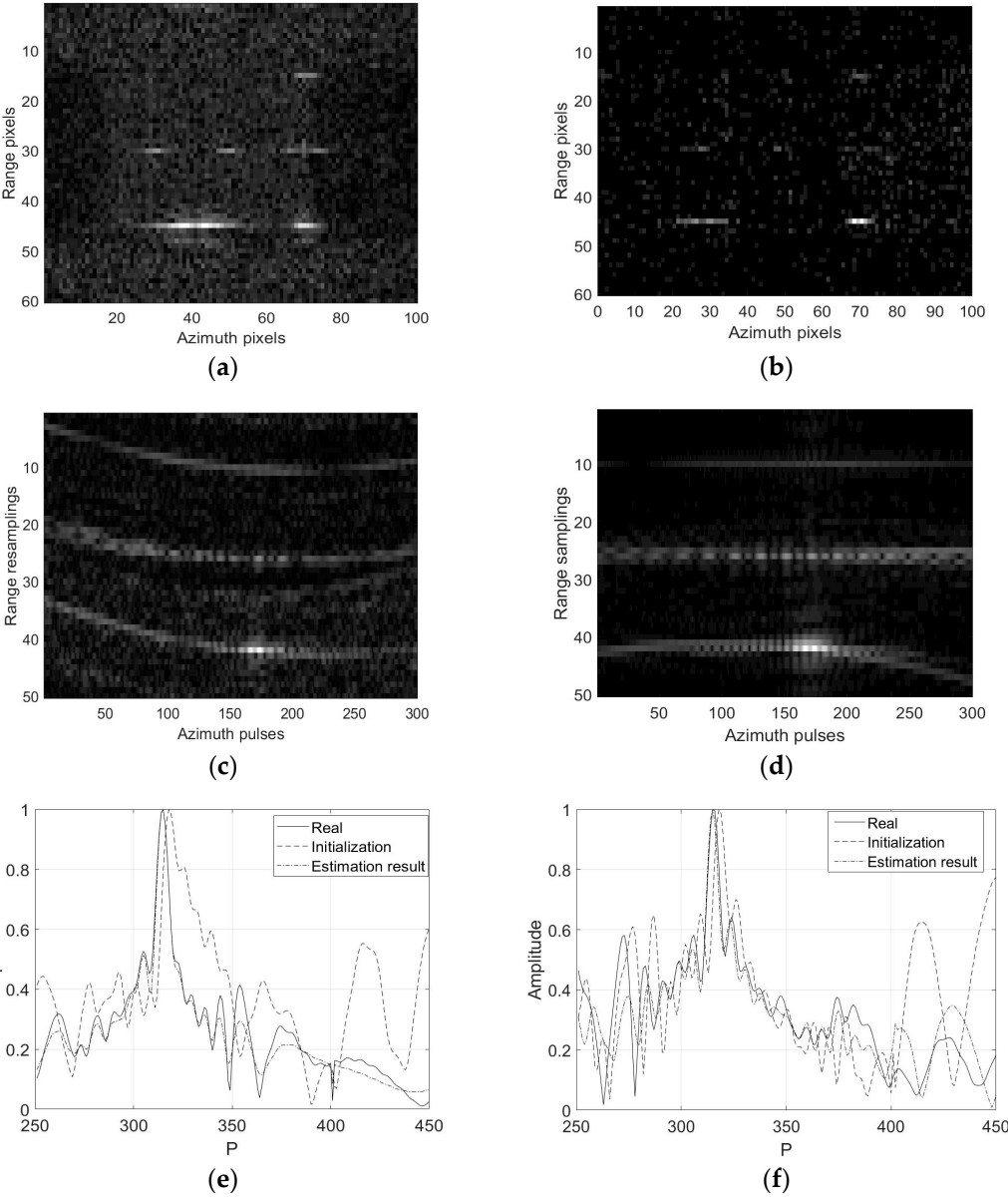

**Figure 5.** Initialization and Doppler rate estimation results. (**a**) initialized MT imagery, (**b**) reconstructed MT imagery after the first iteration, (**c**) reconstructed MT radar data, (**d**) reconstructed MT data after RCMC, (**e**) Doppler estimation results of MT1, and (**f**) Doppler estimation results of MT5.

To further verify our method, the azimuth profiles of the five MTs in the SAR images derived using matched filtering, L1 norm regularization, and our method are depicted for comparison in Figure 6. Moreover, point target analysis with 32 times oversampling was performed and the calculated parameters, including the impulse response width (IRW/m), peak side-lobe ratio (PSLR/dB), and the integrated side-lobe ratio (ISLR/dB), are listed in Table 2. Since the smearing of MTs mainly existed along the azimuth dimension, we only gave the azimuth analysis results here. It can be seen from Figure 6 that the amplitude values of the side lobes derived using matched filtering and L1 norm regularization methods were still very large. Furthermore, the higher ISLR values listed in Table 3 also indicate that there were strong residual clutter and static targets energy in the reconstructed images. In contrast, the side-lobe energy and ISLR values of the MTs obtained using our method were decreased significantly, which verified that our method could effectively implement the suppression of clutter.

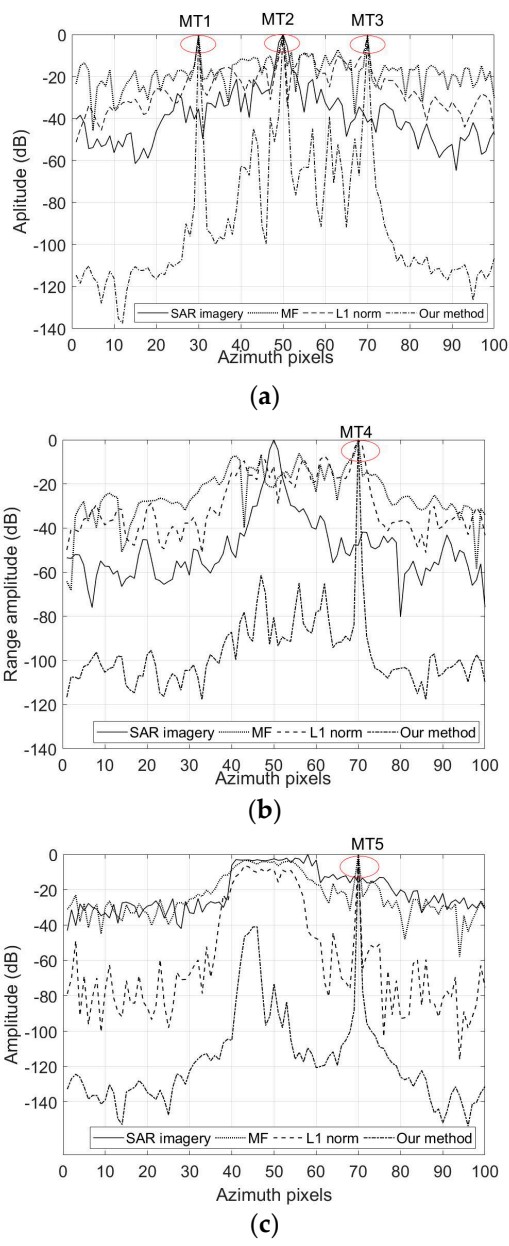

**Figure 6.** Comparison of the azimuth profiles of MTs before and after clutter suppression: (**a**) MT1, MT2, and MT3; (**b**) MT4; and (**c**) MT5.

**Table 3.** Point target analysis results.

| | Frequency Domain Matched Filtering | | | L1 Norm Regularization | | | Our Method | | |
|---|---|---|---|---|---|---|---|---|---|
| | IRW | PSLR | ISLR | IRW | PSLR | ISLR | IRW | PSLR | ISLR |
| MT1 | 0.46 | −7.949 | 0.262 | 0.31 | −12.826 | −8.463 | 0.26 | −13.639 | −9.772 |
| MT2 | 0.36 | −6.065 | −1.338 | 0.31 | −12.615 | −8.287 | 0.27 | −13.028 | −10.264 |
| MT3 | 0.35 | −6.623 | 1.071 | 0.31 | −14.200 | −7.976 | 0.25 | −13.941 | −9.7463 |
| MT4 | 0.31 | −10.814 | 1.218 | 0.29 | −13.078 | −7.318 | 0.26 | −13.221 | −9.8352 |
| MT5 | 0.31 | −9.889 | 13.455 | 0.29 | −12.901 | 5.4789 | 0.26 | −13.266 | −9.6123 |

Moreover, the radar signal illuminated from six MTs with velocities of 40 m/s and 1.8 m/s along the azimuth and range dimensions combined with real SAR data that is provided by the Air Force Research Laboratory (AFRL) was processed using our method. The scene imagery is given in Figure 7a, where the white rectangular frames indicate the position of the defocused MTs. The patch decomposition operation was employed on the original radar data, where the obtained sub-imagery and radar patch data are given in Figure 7b,c, respectively. In these two figures, most of the energy of the MTs is masked by the clutter. The joint sparse-based processing was then used to process the radar patch data with the initialized velocities $v'_x = 45$ m/s and $v'_x = 0$ m/s. The obtained MT imagery based on the initialized velocities is depicted in Figure 7d, in which residual clutter still existed as a result of the utilization of a mismatched radar projection operator. After MT data reconstruction and radar system operator updating, MT imagery and radar data could be derived, as seen in Figure 8c,d and Figure 9c,d. It could be seen that the clutter was significantly suppressed, which further verified the validity of our method.

To assess the performance of our algorithm, we combined the AFRL real SAR data with signals illuminated from 124 MTs and measured the number of successful reconstructions using the traditional constant false alarm rate (CFAR) detection processing. The azimuth velocities and SCR values of the simulated MTs are given in Table 4. Data patch decomposition operations were performed and then sub-images were derived and combined. After joint sparse processing, the MT imagery could be obtained and the traditional CFAR detection operation was performed to measure the performance of the algorithm. The detection results are depicted in Figure 10b and given in Table 4, where the false detected targets are marked by the white ellipse and the missed detected targets are denoted with the red rectangles. Since the main purpose of our paper was to implement the refocusing processing of MTs, the detection results were not compared with the other detection algorithms. It can be concluded from Figure 10 and Table 4 that our algorithm could realize the refocus processing of multiple MTs effectively.

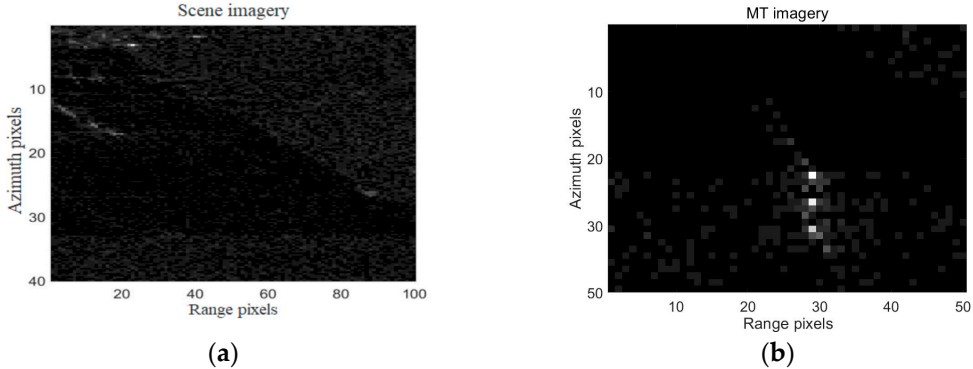

**Figure 7.** MF and sparse imaging result: (**a**) SAR data, (**b**) scene imagery, (**c**) enlarged rectangular area on the left edge of Figure 7a, and (**d**) enlarged rectangular area on the right edge of Figure 7a.

**Figure 8.** *Cont.*

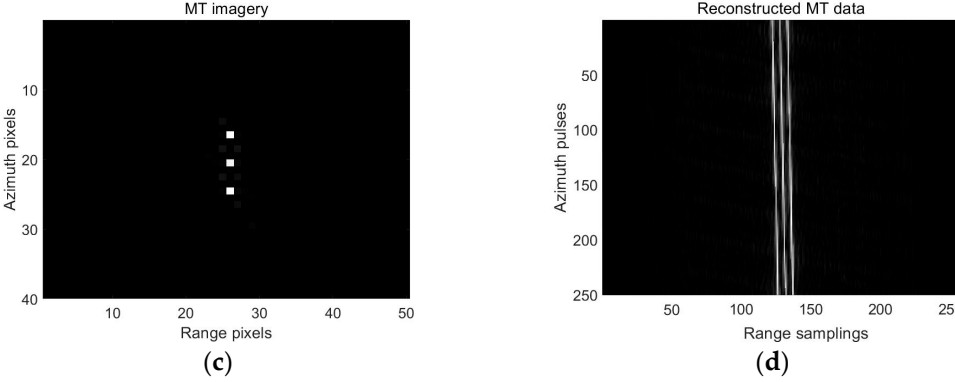

**Figure 8.** Sparse imaging results derived from the defocused patch in Figure 7c: (**a**) reconstructed static scene imagery, (**b**) reconstructed MT imagery after the first iteration, and (**c**,**d**) reconstructed MT imagery and radar data.

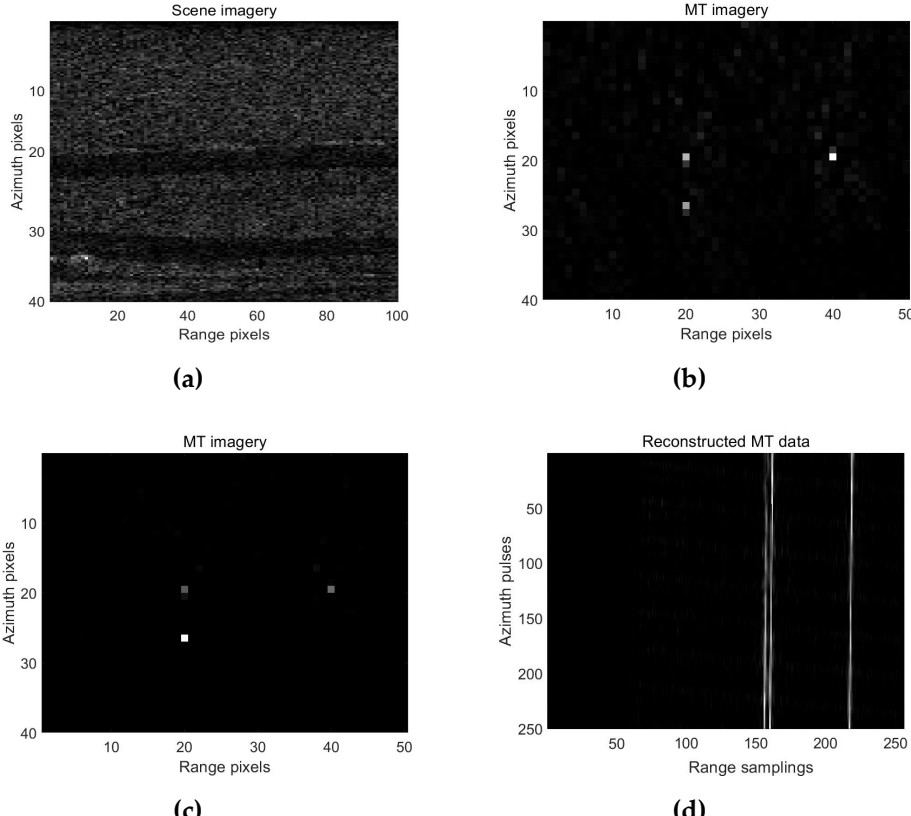

**Figure 9.** Sparse imaging results derived from the defocused patch in Figure 7d: (**a**) reconstructed static scene imagery, (**b**) reconstructed MT imagery after the first iteration, and (**c**,**d**) reconstructed MT imagery and radar data.

**Table 4.** Detection results on the reconstructed MT imagery.

|  | Successful Detection | False Detection |
|---|---|---|
| 19 MTs, 10 dB–20 dB, $v_x = 20$ m/s | 19 | 0 |
| 19 MTs, 10 dB–20 dB, $v_x = 25$ m/s | 19 | 0 |
| 23 MTs, 0 dB–10 dB, $v_x = 30$ m/s | 23 | 0 |
| 23 MTs, 0 dB–10 dB, $v_x = 35$ m/s | 24 | 0 |
| 20 MTs, −8 dB to 0 dB, $v_x = 40$ m/s | 19 | 2 |
| 20 MTs, −8 dB to 0 dB, $v_x = 45$ m/s | 20 | 3 |

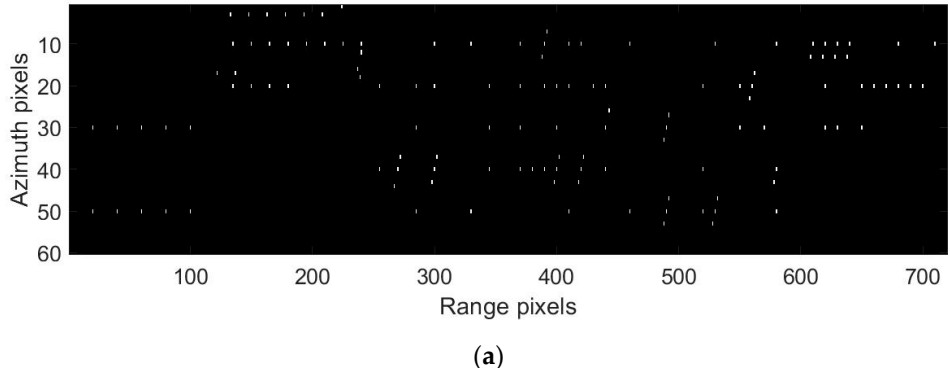

(**a**)

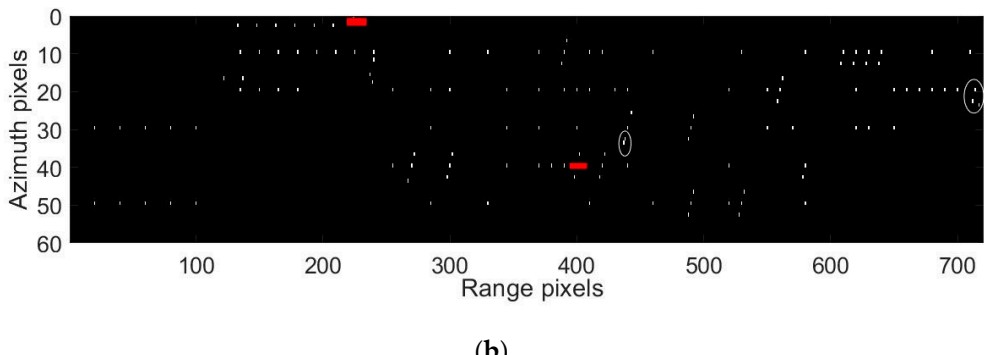

(**b**)

**Figure 10.** MTs and detection results after joint sparse-based refocusing: (**a**) simulated MTs, and (**b**) detection results.

## 5. Conclusions

An MT refocusing method based on the joint sparse imaging is proposed in this paper. The defocused ROI in the SAR imagery was utilized in this method and hence we did not need to process the whole illuminated scene and original radar data. Numerical simulations verified that the presented methodology could realize multiple moving target refocusing effectively.

**Author Contributions:** Conceptualization, methodology, formal analysis, and writing: X.W.; investigation: L.Q.

**Funding:** This research was funded by the Natural Science Foundation of Jiangsu Province (BK20160915), the National Natural Science Foundation of China (61801232), and the National Natural Science Foundation of China (61601240).

**Conflicts of Interest:** The authors declare no conflict of interest.

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
