# Peer review of "Refocusing of Multiple Moving Targets Based on the Joint Sparse Processing of One Channel Synthetic Aperture Radar Imagery Patches"

_electronics, doi:10.3390/electronics8111215_

Round 1

Reviewer 1 Report

This paper is a revised manuscript for the novel focusing method for moving target in SAR system.

The revised manuscript seems fine. However, the numerical evaluations are insufficient. The authors showed only qualitative results and no quantitative results.

Generally, the analysis over a point target is done with oversampled (e.g., 32 or 64 times over sampled) images. Numerically, point targets are evaluated with, for example, geometric accuracy, resolution, and peak-to-sidelobe ratio.

Author Response

We thank for the Reviewer’s suggestion that reminds us to modify our simulation section. We added the point target analysis results in Table III in the revised paper.

Reviewer 2 Report

See attached file

Author Response

We thank for the reviewers’ comments again. Those comments are all valuable and helpful for revising and improving our paper. We have studied the comments carefully and have made modifications. The revised portion are marked in red in the paper. The main corrections in the paper and the responses to the reviewer’s comments are as flowing.

Reviewer 2's Comments:

This paper is technically correct and, in my personal opinion, major revision is required.

Figure 1 is not clear, please insert the range, azimuth and height coordinates directions.

Response:

We follow the reviewer’s suggestion and insert the range, azimuth and height coordinates directions in Figure 1.

Introduction:

2.1 The basics of the Compressed Sensing theory are not cited. Please insert the mother papers of Emmanuel Candes and Terence Tao. The basics of Compressed Sensing radar are not cited, please insert papers concerning the following issue: “Compressed Sensing radar”.

Response:

   We agree with the Reviewer’s suggestion and insert some references concerning “Compressed Sensing radar” in the introduction section.

2.2 SAR image are not sparse, please highlight the strategy you have used in order to sparsify the SAR image. Give an assessment on the probabilities of: Detection, False alarm, Miss and Correct rejection, also on low-energy targets.

Response:

We highlight the sparse strategy in the introduction and algorithm sections and consider the reviewer’s comment giving the assessment on the probabilities of Detection and False alarm carefully. The main purpose of our algorithm is to realize the refocusing processing of MTs, so we do not discuss the detection probability and false alarm on the original paper. To further verify our method, we follow the reviewer’s comment and perform the CFAR detection to verify that multiple MTs can be reconstructed in the imagery derived via our method. The numbers of success full detection and false detection are listed in Table IV in the revised paper to prove the validity of our algorithm.

Figures 4 and 5, use grid on.

Response:

   We follow the reviewer’s suggestion and use the grid in the two figures.

Round 2

Reviewer 2 Report

accepted

This manuscript is a resubmission of an earlier submission. The following is a list of the peer review reports and author responses from that submission.

Round 1

Reviewer 1 Report

The authors proposed a novel moving target refocusing method for SAR imagery. The proposal is reasonable however; their experiments seem insufficient.

1. What is the reason of the parameter definitions in the Experiments? If the authors want to put multiple targets, it is strongly advised to set specific parameters with good reasons to show the accuracy and robustness of the proposed method. Targets have to represent typical motion such as range/azimuth motion only, appearing at the similar place with different speed and positions, etc.

2. The authors have to compare the proposed method with conventional methods precisely. There have been multiple methods to deal with moving targets in SAR imagery. Please clarify the superiority, cons and pros.

3. Some references are cited with smaller fonts (e.g., 1-16, 19-22) while the others are with regular font size (17, 18). Please follow the format.

4. Table 1 does not provide enough information. There must be parameters for SAR platform. Speed and direction of moving targets should be summarized in somewhere.

Reviewer 2 Report

This paper describes a Synthetic aperture radar (SAR) methodology for tracking multiple moving targets through clutter regions.  

This appears to be a reasonable piece of work in the area of radar imaging; however, this makes it wholly unsuited to this journal. It is heavily jargon-filled as the topic is very esoteric, which does not suit the broader audience of the journal ‘Electronics’. It would be better suited to, for example, ‘Remote Sensing’, and the authors should consider re-submission there.

A quick sweep on the topic reveals similar papers in the area [1]–[4]. These should be investigated and differentiated from this work before re-submission in a more appropriate journal.

[1] Y. Chen, G. Li, Q. Zhang, and J. Sun, “Refocusing of moving targets in SAR images via parametric sparse representation,” Remote Sens., vol. 9, no. 8, pp. 1–15, 2017.

[2] Y. Zhang, J. Sun, P. Lei, G. Li, and W. Hong, “High-Resolution SAR-Based Ground Moving Target Imaging With Defocused ROI Data,” IEEE Trans. Geosci. Remote Sens., vol. 54, no. 2, pp. 1062–1073, 2016.

[3] G. Li, Y. Chen, and Q. Zhang, “Iterative Minimum-Entropy Algorithm for Refocusing of Moving Targets in SAR Images,” IET Radar, Sonar Navig., 2019.

[4] M. Çetin, M. Yasin, and A. S. Khwaja, “A subaperture based approach for SAR moving target imaging by low-rank and sparse decomposition,” in Algorithms for Synthetic Aperture Radar Imagery XXV, 2018, vol. 10647, p. 19.